# The Correlation between Peripheral Blood Index and Immune Cell Expansion in Vietnamese Elderly Lung Cancer Patients

**DOI:** 10.3390/ijms24054284

**Published:** 2023-02-21

**Authors:** Hoang-Phuong Nguyen, Viet Anh Bui, Ai-Xuan Thi Hoang, Phong Van Nguyen, Dac-Tu Nguyen, Hien Thi Mai, Hai-Anh Le, Thanh-Luan Nguyen, Nhung Thi My Hoang, Liem Thanh Nguyen, Xuan-Hung Nguyen

**Affiliations:** 1Vinmec Institute of Stem Cell and Gene Technology, Vinmec Healthcare System, 458 Minh Khai, Hanoi 100000, Vietnam; 2Center of Applied Science, Regenerative Medicine, and Advance Technologies (CARA), Vinmec Healthcare System, 458 Minh Khai, Hanoi 100000, Vietnam; 3Vinmec Times City International Hospital, Vinmec Healthcare System, 458 Minh Khai, Hanoi 100000, Vietnam; 4Faculty of Biology, VNU University of Science, 334 Nguyen Trai Street, Hanoi 100000, Vietnam; 5College of Health Sciences, VinUniversity, Hanoi 100000, Vietnam

**Keywords:** immune cell expansion, immunosenescence, peripheral blood indices, CD56bright NK cells, cancer, aging

## Abstract

(1) Background: The dysfunction and reduced proliferation of peripheral CD8^+^ T cells and natural killer (NK) cells have been observed in both aging and cancer patients, thereby challenging the adoption of immune cell therapy in these subjects. In this study, we evaluated the growth of these lymphocytes in elderly cancer patients and the correlation of peripheral blood (PB) indices to their expansion. (2) Method: This retrospective study included 15 lung cancer patients who underwent autologous NK cell and CD8^+^ T cell therapy between January 2016 and December 2019 and 10 healthy individuals. (3) Results: On average, CD8^+^ T lymphocytes and NK cells were able to be expanded about 500 times from the PB of elderly lung cancer subjects. Particularly, 95% of the expanded NK cells highly expressed the CD56 marker. The expansion of CD8^+^ T cells was inversely associated with the CD4^+^:CD8^+^ ratio and the frequency of PB-CD4^+^ T cells in PB. Likewise, the expansion of NK cells was inversely correlated with the frequency of PB-lymphocytes and the number of PB-CD8^+^ T cells. The growth of CD8^+^ T cells and NK cells was also inversely correlated with the percentage and number of PB-NK cells. (4) Conclusion: PB indices are intrinsically tied to immune cell health and could be leveraged to determine CD8 T and NK cell proliferation capacity for immune therapies in lung cancer patients.

## 1. Introduction

Lung cancer is the most prevalent type of cancer and is responsible for the highest cancer-associated mortality rate worldwide, as well as in Vietnam [1]. However, there is still a lack of efficient therapy for treating this disease, especially for patients with late stages of lung cancer. In the last decade, CD8^+^ T and NK cells have gained much attention for their potent cytotoxic capacity and ability to act as a living drug to treat cancer. These cells can be collected from peripheral blood (PB), expanded ex vivo, and administered back to the patients. Such immune cell therapies can inhibit tumor growth by boosting the immune response and have shown success in several advanced cancers [2,3,4,5,6].

Natural killer (NK) cells are an essential component of the innate immune system to eliminate virally infected and transformed cells. Recently, the role of NK cells has been revealed extensively in antimicrobial response [7,8], senescent cell elimination [9], inflammatory resolution [10,11], and induction of adaptive immune responses. An 11-year follow-up study involving 3625 people aged 40 and over in Japan indicated an association between lower NK cell activity and increased cancer risk [12]. T cells also play a central role in cancer immune responses by specifically recognizing and reacting to tumor-expressing antigens and have been critically deployed for cancer immunotherapy [13]. Immune cells need to engage directly with tumor cells to perform lethal attacks; thus, higher levels of intratumoral lymphocyte infiltration are also associated with better overall survival in cancer patients [14,15].

One of the biggest challenges to the success of immune cell therapy is the efficient expansion of immune cells of interest, especially in elderly patients characterized by immunosenescence and the reduction of the proliferation rate of both NK cells and T cells [16,17,18,19]. Additionally, T cells in cancer patients are exhausted rapidly due to chronic tumor antigen exposure. These dysfunctional T cells are characterized by low proliferative potential, poor effector function, and overexpression of multiple inhibitory receptors [13,18,20]. Different methods have been developed to restore the function of these dysfunctional immune cells for therapeutic efficiency enhancement [21]. Human NK cells are also not homogenous and can be divided into four different subsets based on the expression of CD16 and CD56 expression. Although accounting for a small fraction, about 10% of circulating NK cells, CD56^bright^ NK cells are superior at producing pro-inflammatory cytokines and act as immunomodulatory cells [22]. CD56^dim^ NK cells are the major blood NK subset (up to 90%) and are potent mediators of natural and antibody-dependent cytotoxicity [22,23,24,25]. The ratio of CD56^bright^:CD56^dim^ is impaired due to aging and the emergence of cancer, and the circulating CD56bright NK cells inversely correlate with the survival of cancer patients [26,27,28]. Thus, the current NK cell expansion technologies for cancer immune cell therapy are focusing on the expansion of this CD56^bright^ NK cell subpopulation.

Complete lymphocytic blood cell count reflects the peripheral immune status of cancer patients. A recent study indicated that absolute lymphocytic counts of preoperative breast cancer patients could predict disease outcomes [29]. In patients with extensive small-cell lung cancer, the composition of circulating T lymphocyte subtypes is an alternative marker of tumor progression [30]. Besides, immune cell profiling of PB served as a signature upon immune checkpoint blockade across different types of cancer [31,32]. However, there is little evidence of the relationship between the composition and expansion capacity of the PB lymphocytes.

In this study, we investigated the proliferation of circulating CD8^+^ T cells and NK cells from cancer patients of different genders, ages, and cancer types. We then investigated the correlation between the composition/frequency of circulating lymphocytes and the expansion capacity of immune cells in elderly cancer subjects.

## 2. Results

### 2.1. Patient Characteristics

We collected peripheral blood mononuclear cells (PBMCs) from 15 patients (4 males, 11 females) diagnosed with lung cancer and 10 healthy individuals (6 males, 4 females) as a control group. The mean age was 65.2 ± 7.37 and 59.0 ± 12.4 years old in the control and lung patient groups, respectively. The patient cohort comprised 4 patients at stage III, and 11 patients at stage IV. The duration of autologous immune enhancement therapy (AIET) after surgery or the 1st chemotherapy was 16.2 months (ranging from 2 to 48 months). The estimated survival time prior to AIET was 12 months (ranging from 3 to 20 months) (Table 1).

### 2.2. The Peripheral Blood Indexes

First, we characterized the absolute number of WBC and lymphocytes and the relative frequencies of the CD3^+^, CD4^+^, and CD8^+^ T cells and NK cells in both lung cancer patients and healthy individuals. In lung cancer patients, the median density values of WBC and lymphocytes, the frequency of lymphocytes and CD3^+^ T, CD4^+^ T, and CD8^+^ T cells, were at the lower boundary of the normal range, and lower than those in the healthy individuals, but not significantly different (*p* < 0.05) (Figure 1). The median frequency of NK cells in lung cancer patients was within the normal range and higher than that of healthy people but not significantly different (*p* > 0.05) (Figure 1).

However, a large portion of patients’ blood indices was out of the normal range, in which the frequency of CD4^+^ T cells, CD3^+^ T, and lymphocytes in 60% (9/15), 53.3% (8/15) and 40% (6/15) of the patients, respectively was out of range. In healthy people, blood indices were less likely to be out-of-range compared to lung cancer patients (Figure 1H).

Taken together, these findings demonstrated that the PB indices of lung cancer patients enrolled in the study had the tendency to be on the lower end of the spectrum and that they had a high rate of out-of-range values.

### 2.3. Immune Cell Expansion Capacity

To evaluate the capacity of peripheral immune cell expansion in cancer patients, the PBMCs were collected and cultured using BINKIT [4]. From day 3 (D3) onward, the total cell number was counted every two days. CD8^+^ T lymphocyte culture reached the log phase earlier than NK cell culture (D5-7 vs. D7-9, respectively).

From D0 to D3, the total cell numbers in CD8^+^ T and NK cell cultures decreased by 66.7% (10/15) and 100% (15/15) in lung cancer samples, and 60% (6/10) and 80% (8/10) in healthy samples, respectively. From D5, the cell counts more than doubled every two days, with a peak fold-increase at D7 in CD8^+^ T cell culture (3.2 times for both cancer and healthy samples) and D11 in NK cell culture (2.9 times in cancer and 2.7 times in healthy samples), followed by a gradual decline until D15 (Figure 2A1, A2). After 15 days, CD8^+^ T cell culture reached a higher total cell count than NK cell culture (Figure 2B1,B2).

To determine the number of CD8^+^ T cells and NK cells, the samples were analyzed by flow cytometry on D0 and D15 (Figure 2C).

In lung cancer samples, after 15 days of culture, the number of CD8^+^ T lymphocytes increased 405 (108–2510)-fold to 3313.0 ± 2382.5 × 10^6^ cells, with a cell viability of 97.6% and frequency of 68.7 ± 14.9%. Meanwhile, the number of NK cells increased 147.4 (22–2495)-fold to 2320.1 ± 2504.9 × 10^6^ cells, with a cell viability of 97.5% and frequency of 79.1 ± 23.8%. In healthy samples, the fold increase of CD8^+^ T cells and NK cells was 783 (384–5763) and 2150 (547–5664), respectively. The expansion capacity of NK cells in healthy people was significantly higher than that of lung cancer patients (Figure 2D1,D2).

After 15 days of culture, the average frequency of CD56^bright^ NK cells after the expansion increased from 2.1 ± 1.4% to 96.9 ± 3.5% in lung cancer patients, which increased 46.7 times. Meanwhile, in healthy people, the frequency of CD56^bright^ NK cells in PB and after expansion was 0.19 ± 0.1% and 85±11.9%, respectively, which were both lower than in the lung cancer samples (*p* < 0.01) (Figure 2E).

Together, we demonstrated that both the number and purity of PB-CD8^+^ T cells and PB-NK cells significantly increased after 15 days of culture. However, there was still a large variation in the expansion capacity between samples.

Because the peripheral CD8^+^ T cells and NK cells from several patients exhibited limited proliferation capacity, we analyzed the impact of age, gender, metastasis, and cancer stages on the expansion of these cell types. In the PB, the density of lymphocytes and the frequencies of CD3^+^CD8^+^ T cells were likely higher in females than in males, although there was a gender imbalance in our cohort (*p* < 0.05). However, the number and the fold increase of both CD8^+^ T cells and NK cells after expansion were not significantly different by gender (*p* > 0.05). In addition, the density of PB lymphocytes in metastasis patients was significantly higher than in non-metastasis subjects (*p* < 0.05), but the expansion capacity was not significantly different between the two groups (*p* > 0.05). Age, dividing by above and below 60 years, and cancer stages (III and IV) had no impact on the immune cell expansion (*p* > 0.05). Interestingly, the expansion of NK cells was significantly higher in healthy people than in lung cancer patients (*p* < 0.001) (Figure 3). Collectively, this result demonstrated that CD8 T cell and NK cell expansion capacity was not affected by sex, age, metastasis state, and cancer stages. However, cancer status does affect NK cell expansion.

### 2.4. The Immune Cell Expansion in Out-of-Range Samples

We have shown that a large portion of lung cancer subjects has out-of-range PB indices. Here, we investigated whether the expansion capacity is related to the range of PB index values.

In the patients with the lower frequency and density of PB-lymphocytes (<16.8% and <900 cells/µL, respectively), we observed a decrease in the frequency of mononuclear cells (*p* < 0.01), the number of PB-CD8^+^ T cells (*p* < 0.01), and PB-NK cells (*p* < 0.05) compared to that in the patients within the normal range (Figure 4A,B). Surprisingly, patients with a lower-than-normal frequency of PB-CD8 T cells and PB-NK cells had no significant difference in either PB or post-culture indices (Figure 4C,D).

These results indicated that the out-of-range value of examined samples was not correlated with the expansion of immune cells. We further investigated the relationship between the PB indices and the immune cell expansion capacity.

### 2.5. The Relationship between the PB Indices and the Immune Cell Expansion

We next investigated the correlation between PB indices and the expansion capacity of immune cells. The frequency of expanded CD8^+^ T cells was positively correlated with the PB-CD8^+^ T cell frequency (r = 0.57) and negatively correlated with PB-CD4^+^ T cell frequency (r = −0.61) and the CD4:CD8 ratio (r = −0.61). CD8^+^ T cell expansion capacity was also inversely correlated with the percentage and number at seeding of PB-NK cells (r = −0.68, and r = −0.72, respectively) (Figure 5A).

With regards to NK cell expansion, the frequency of PB-lymphocytes, PB-NK cells, and the seeding number of CD8^+^ T cells were negatively correlated with the fold increase of this cell type (r = −0.58, −0.65, and −0.53, respectively). Particularly, the seeding number of NK cells had inverse correlations to the fold increase of the cells (r = −0.61) (*p* < 0.05) (Figure 5B). Interestingly, CD56^bright^ NK cell expansion capacity also had a strong correlation with PB-CD56^bright^ NK frequency (r = 0.99, *p* = <0.0001) (Figure 5C). On the other hand, we found no correlation between the number of CD8^+^ T cells or NK cells and PB indices (*p* > 0.05).

Remarkably, in healthy people, there was no correlation between immune cell expansion capacity and the PB indices, except for the tight relationship between the fold increase of CD8 T cells and the frequency of PB-NK cells (r = 0.66, *p* = 0.043) (Figure 6).

In summary, our analysis showed that PB indices had a robust relationship with immune cell expansion. Expansion of CD8 T cells, NK cells, and CD56^bright^ NK cells are correlated with PB-NK cell number, PB-lymphocyte percentage, and PB-CD56^bright^ NK frequency, respectively.

## 3. Discussion

Immune cell expansion from PB lymphocytes of elderly cancer patients is challenging because of the immunosenescence caused by aging and cancer [16,17,18,19,20]. In this study, we aimed to identify that the PB indices in elderly lung cancer patients were correlated with CD8^+^ T lymphocytes and NK cell expansion. Our results showed that more than 35.9% of lung cancer patients have a number and frequency of PB lymphocytes lower than the reference range for healthy people of the same age, which is consistent with the previous studies reporting the decline in the lymphocyte count in cancer patients [20,33,34,35]. Particularly, 60% and 53.3% of patients experienced a decrease in the frequency of PB-CD4^+^ T cells and PB-CD3^+^ T cells, respectively. Several studies indicated an association between low absolute lymphocyte count and worse disease-free survival and higher mortality in cancer patients [36,37,38,39,40,41]. The circulating CD4^+^ T cells are significantly decreased in non-small cell lung cancer and small lung cancer patients [19,42]. In contrast to the observed reduction in CD4^+^ T cells, the frequency of total peripheral blood NK cells did not decrease in lung cancer patients. This result was consistent with previous observations that the number and percentage of NK cells did not decline in cancer patients [16,17]. In addition, the T cells can become dysfunctional, characterized by reduced proliferative capacity [18,20]. Thus, restoring the peripheral lymphocytes to the normal range could be beneficial to cancer treatment efficacy, especially in elderly subjects.

We addressed whether it is possible to expand CD8^+^ T cells and NK cells in the patients and whether the PB lymphocytic cell count is associated with this expansion. After 15 days of culture, both CD8^+^ T and NK cells had expanded fold changes ranging from 100 to 2500 times, which is consistent with other reports with the same culture period [43,44,45,46]. However, the growth of NK cells significantly declined compared to healthy people, similar to the report of Gounder SS et al. [16]. Although several clinical trials showed low NK cell expansion in lung cancer patients [6,43,44,47], our culture method was effective in inducing the proliferation of these lymphocytes, even in elderly patients with an average age of 59.2 years. We observed that the optimal time for cell growth was from day 7 to day 11 of the culture. The extended culture might lead to cell senescence, as indicated by the decreased expansion capacity after 13 days despite the cell number continuing to increase [45,46]. In our study, the frequency of the CD56^bright^ NK cell subset in the expanded NK cell population was higher than 95%. CD56^bright^ NK cells were reported to decline with age in elderly subjects [16,17,48]. Our culture method successfully expanded this NK cell subpopulation, thereby suggesting the functional restoration of amplified NK cells. Moreover, the BINKIT used in this study could expand not only CD3-CD56 NK cells but also CD3CD56 NKT cells. This cell population also plays an important function in targeting and killing cancer cells and has been used in clinical trials for cancer treatment [49,50].

In accordance with previous studies suggesting the impact of gender on immune cell proliferation [16,17,18,19,20], we observed that the higher expansion capacity of CD8^+^ T cells in females was higher than that in males. However, the metastasis state and cancer stages had no impact on the growth of these two immune cell types in our study.

Several studies suggested that the total number of NK cells and T cells increased with aging, accompanied by the expansion of exhausted or senescent cells [16,19,49,50], leading to a decrease in peripheral immune cell proliferation. We observed that a high frequency of PB-CD8^+^ T cells and NK cells had a contrary effect on the fold increase of the two immune cell types, which increased in CD8^+^ T cells but decreased in NK cell cultures. Interestingly, a smaller number of NK cells at seeding correlated to a better effect in cell expansion. This may be due to the higher frequency of exhausted CD56^dim^ NK cells in the PB-NK population of lung cancer patients. The expansion of NK cells was also inversely correlated with the frequency of PB-lymphocytes and the density of PB-CD8 T cells.

Our study proposed that the frequency of PB-CD4^+^ T cells and the CD4:CD8 ratio was inversely proportional to the frequency of the CD8^+^ T cells in the expanded population. The percentage and concentration of PB-NK cells also negatively correlated to the expansion of CD8^+^ T cells. Our results pose the possibility that a decrease in the total lymphocytes might not be detrimental if it is accompanied by a decrease in the number of senescence PB-NK or PB-CD8 T cells. Additional studies are needed to evaluate the correlation between immune cell expansion, cancer stage, and the expression of exhausted immune markers. Particularly, the correlation between PB-NK frequency and count to the immune cell expansion was not observed in healthy donors. That means these correlations are likely related to lung cancer patients. However, because of the small cohort in this study, more samples should be collected in future work to get confirmation of this relationship.

Our study also has several limitations. First, the impact of previous treatments, such as chemotherapy and radiotherapy, was not investigated, which can affect the number of circulating lymphocytes. Second, the number of patients was limited to 15, reducing the statistical power of comparing PB indices between young and old cancer patients. Third, gender is imbalanced in the patient cohort. Unfortunately, this is a retrospective study, so it is difficult to recruit more patients. Therefore, we have anticipated this in the manuscript and focused the analysis on the whole cohort. However, our findings provided valuable insights that serve as the motivation to delve deeper into predicting and optimizing immune cell expansion for cell therapy.

## 4. Materials and Methods

### 4.1. Study Population

In this retrospective study, we collected data from 15 lung cancer patients who underwent autologous NK cell and CD8^+^ T cell therapy at the Vinmec Times City International Hospital (Hanoi, Vietnam) between January 2016 and December 2019. The study was compiled with the standards of the Helsinki Declaration and approved by the institutional review board of Vinmec International Hospital (approval number: 28/2022/CN-HDDD VMEC).

### 4.2. Data Collection

The clinicopathologic information was extracted from the patients’ medical records, including their gender, age, cancer stages, and metastasis state (Table 1). Complete PB blood cell counts, including white blood cell count (WBC, cells/µL), frequency (%), and absolute numbers (cells/µL) of lymphocytes, frequency (%) of monocytes, and mononuclear cells, were extracted from the blood routine test on the day of blood collection for the first immune cell expansion. The blood routine test was performed by a Semi-Automatic Nihon Kohden MEK9100 Hematology Analyzer (BIONS Medicals Systems Pvt. Ltd. Kochi, Kerala, India).

The proportion of CD3^+^, CD4^+^ T, CD8^+^ T, and NK cells in the peripheral blood lymphocyte population and the number of CD8^+^ T cells and NK cells were determined using flow cytometry.

### 4.3. Immune Cell Expansion

PB samples were collected from patients and transferred to 15 mL polypropylene tubes containing 10% ethylenediaminetetraacetic acid as an anticoagulant. An amount of 2 mL of blood was set aside for counting the total cell number. Approximately 50 mL of blood was used for immune cell expansion for one sitting as described in our previous study [51]. Briefly, PB mononuclear cells (PBMCs) were obtained by density gradient centrifugation using Ficoll-Paque (GE Healthcare, Uppsala, Sweden) and cultured using BINKIT^®^ (Biotherapy Institute of Japan, Tokyo, Japan). The cell culture period was divided into two stages: the initial culture (for the targeting of a specific cell type) and the subculture (for the targeted cell proliferation) [4,46]. The cell processing center was set up in compliance with Good Manufacturing Practice (GMP) standards. In patients treated with radio or chemotherapy, the blood had to be collected one month after the last chemotherapy regimen or before the start of radiotherapy.

The phenotype of expanded cells and PBMCs at baseline (Day 0) and the end of the culture was analyzed by flow cytometry. The following monoclonal antibodies were used: CD3-Pacific Blue, CD8-fluorescein isothiocyanate (FITC), CD56-R Phycoerythrin (PE), and CD4-Allophycocyanin (APC)-Alexa Flour 750, and the corresponding isotypes. All antibodies and isotypes were purchased from Beckman Coulter, CA, USA. All samples were acquired on a Navios Cytometer (Beckman Coulter, Brea, CA, USA). Data were analyzed with the Navios software, version 3.2, according to the manufacturer’s instructions.

### 4.4. Statistical Analysis

Descriptive statistics include frequency, mean, and standard deviation to describe research subjects’ characteristics of cells. Statistical analysis of the frequencies of immune cell subpopulations between groups was performed using the ANOVA with Tukey HSD tests and the two-tailed Wilcoxon rank-sum test with R Program, followed by the Bonferroni correction (Version 1.2.5042). Spearman’s rank coefficient was used for correlation analysis. GraphPad Prism (Version 8.4.3) was used to create grouped box-and-whisker graphs. Results with *p*-values < 0.05 were considered statistically significant.

## 5. Conclusions

In conclusion, we successfully expanded CD8^+^ T lymphocytes and NK cells from the PB of the elderly lung cancer subjects. The expanded NK cells had a high frequency of the CD56^bright^ subset, indicating the restored function in this population. Furthermore, the proliferation of these immune cells correlated with the PB indices, in which the expansion of CD8^+^ T cells was inversely associated with the peripheral blood CD4:CD8 ratio and the frequency of PB-CD4^+^ T cells. The expansion of NK cells was negatively correlated with PB-lymphocytes’ frequency and the PB-CD8^+^ T cell count. The growth of both cell types had an inverse correlation to the percentage and number of PB-NK cells. Our study proposed that it is possible to predict the proliferation rate of immune cells in lung cancer patients based on their PB indices for immune cell therapies.

## Figures and Tables

**Figure 1 ijms-24-04284-f001:**
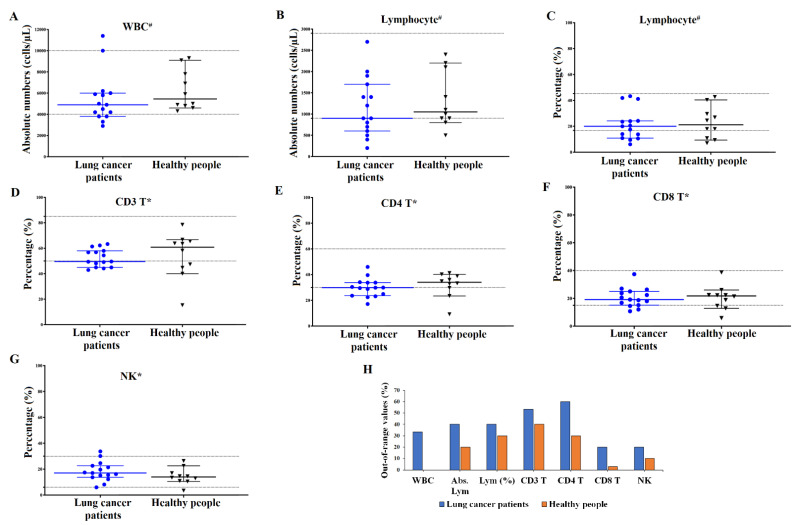
The peripheral blood indices of cancer patients (n = 15) and healthy people (n = 10). The complete PB cells were counted on the hematology analyzer and flow cytometry system. Each data point represents one patient, blue and black lines indicate the median of lung cancer and healthy samples, respectively, and dotted lines indicate the upper and lower limits of reference intervals. (**A**) The absolute numbers of WBCs (cells/µL). (**B**) The absolute numbers of lymphocytes in PB (cells/µL). (**C**) The percentage of peripheral blood lymphocytes (%) among white blood cells. (**D**–**G**) The percentage of CD3^+^ T, CD4^+^ T, CD8^+^ T, and NK cells in the PB lymphocyte population, respectively. (**H**) The frequency (%) of out-of-range values in peripheral blood of lung cancer patients and healthy people. ^#^ Reference from Vinmec hospital. * Reference from Ministry of Health with document number: QĐ1494/2015. Abs.: Absolute numbers.

**Figure 2 ijms-24-04284-f002:**
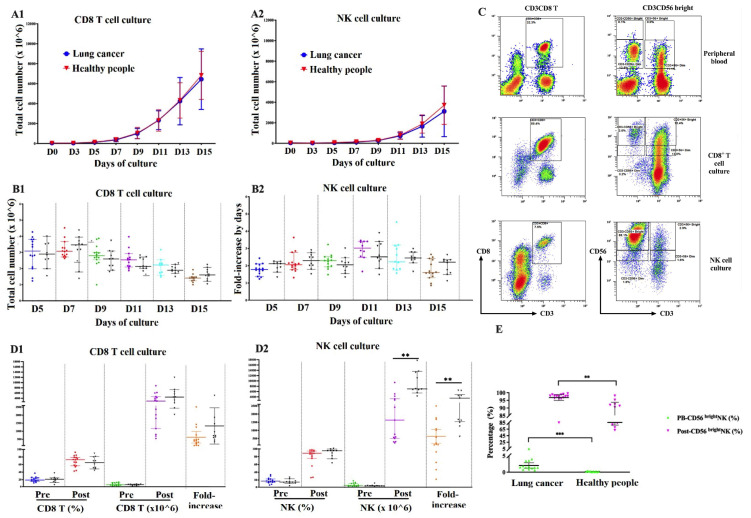
The proliferation of immune cells during 15 days of culture in lung cancer patients and healthy people. The growth curves of total cells in CD8^+^ T cell (**A1**) and NK cell cultures (**A2**). The fold increase of total cells every two days from D3 to D15 in CD8^+^ T cell culture (**B1**) and NK cell cultures (**B2**). The gating strategy to identify and count T cell and NK cell subsets by flow cytometry (**C**). The expansion of NK cells (**D1**) and CD8^+^ T cells (**D2**) after 15 days of culture. The changes in the frequency of CD56^bright^ NK cells in PB and after 15 days of culture (**E**). Pre: at cell seeding; Post: at cell harvesting after culture. Significance levels were set to *p* < 0.01 (**), and *p* < 0.001 (***).

**Figure 3 ijms-24-04284-f003:**
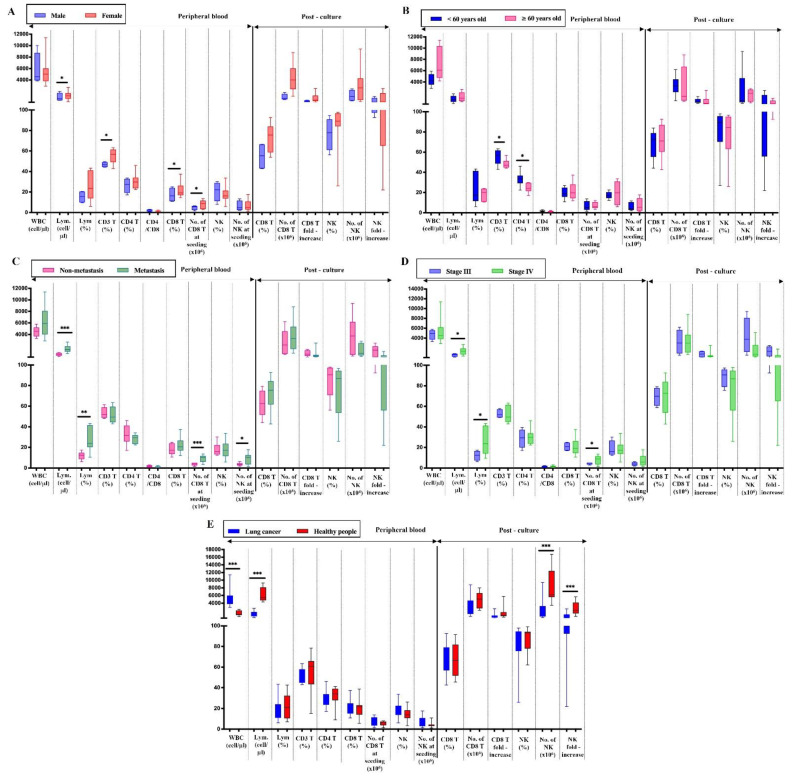
The capacity of immune cell expansion by age, gender, metastasis state, and cancer stages, and between cancer patients and healthy people. The comparison of median values of PB indices and immune cell expansion ability by gender (**A**), age below and above 60 years old (**B**), metastasis state (**C**), and cancer stages III and IV (**D**), and cancer patients and healthy people (**E**). Significant differences were obtained by the Wilcoxon signed-rank test followed by the Bonferroni correction. Median values and interquartile ranges are shown in the graphs. Significance levels were set to *p* < 0.05 (*), *p* < 0.01 (**), and *p* < 0.001 (***). PB: Peripheral blood; Lym.: Lymphocytes. No.: Number.

**Figure 4 ijms-24-04284-f004:**
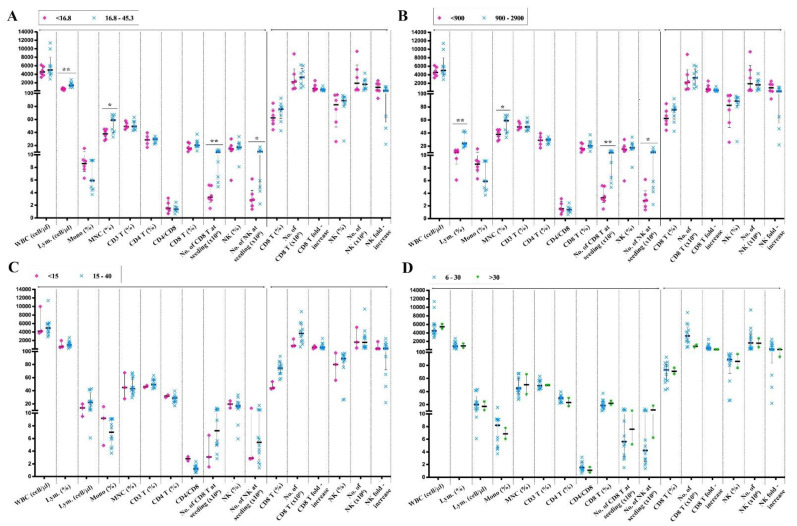
The comparison of immune cell expansion capacity between subjects with a normal and abnormal frequency of PB-lymphocytes (**A**), the density of PB-lymphocytes (**B**), the frequency of PB-CD8^+^ T cells (**C**), and the frequency of PB-NK cells (**D**). Significant differences were obtained by the Wilcoxon signed-rank test followed by the Bonferroni correction. Median values and interquartile ranges are plotted in graphs. Significance levels were set to *p* < 0.05 (*), and *p* < 0.01 (**). PB: Peripheral blood; WBC: whole blood cell count; SL: number of; Lym.: Lymphocytes; MONO.: Monocytes; MNC: mononuclear cells.

**Figure 5 ijms-24-04284-f005:**
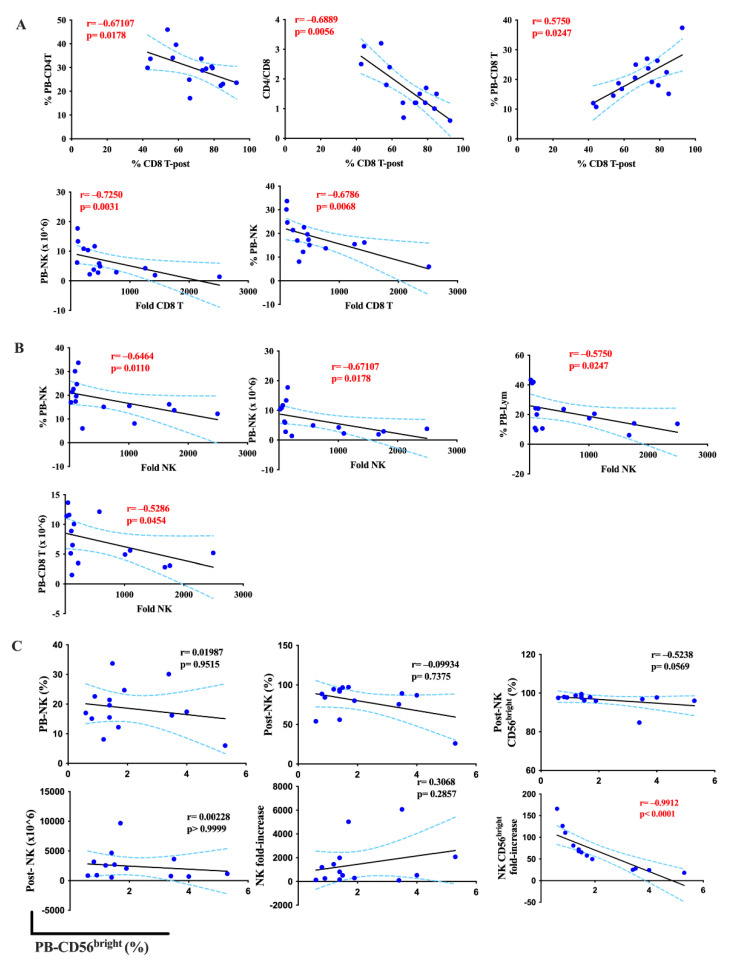
Correlation of peripheral blood indices and the expansion of CD8^+^ T cells and NK cells in lung cancer patients. (**A**) The relationship of PB indices and the expansion capacity of CD8^+^ T cells. (**B**) The relationship between PB indices and the expansion capacity of NK cells. (**C**) Relationship of PB- CD56^bright^ NK (%) with NK cell growth. Spearman’s rank coefficient was used for correlation analysis and non-linear regression was also applied. Median values and interquartile ranges were plotted in graphs. Significance levels were set to *p* < 0.05, *p* < 0.01, and *p* < 0.001. PB: Peripheral blood; Pre-: At seeding; Post-: After expansion.

**Figure 6 ijms-24-04284-f006:**
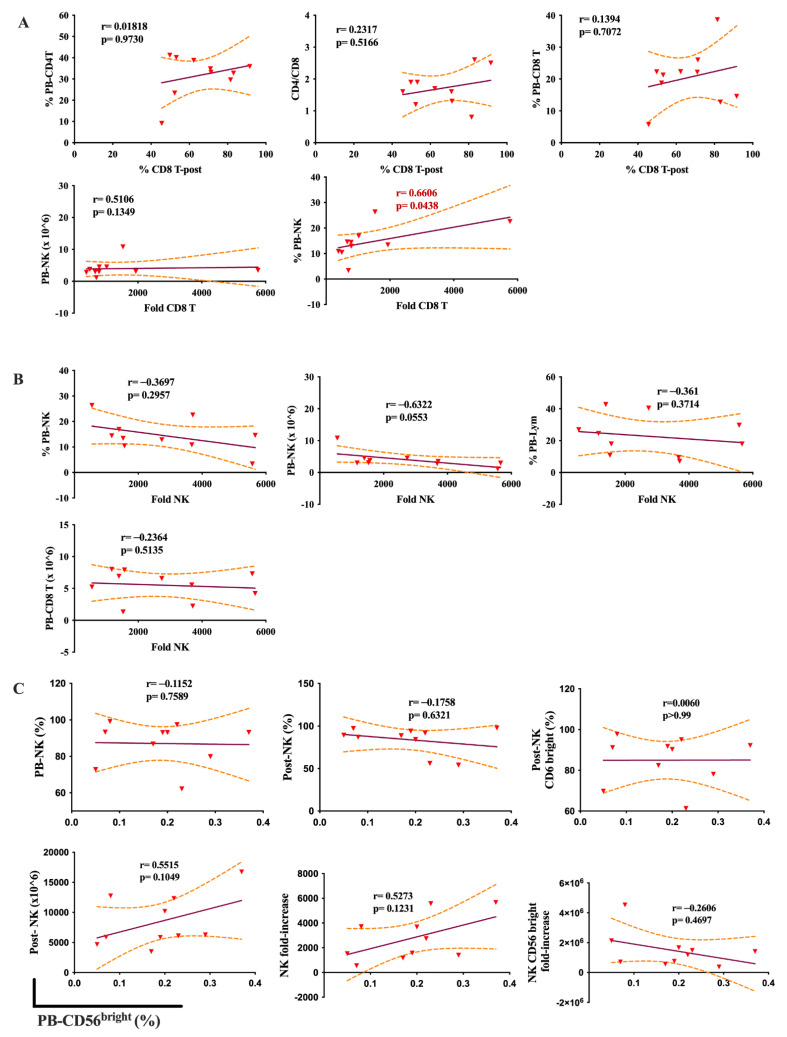
Correlation of peripheral blood indices and the expansion of CD8^+^ T cells and NK cells in healthy donors. (**A**) The relationship of PB indices and the expansion capacity of CD8^+^ T cells. (**B**) The relationship between PB indices and the expansion capacity of NK cells. (**C**) Relationship of PB- CD56^bright^ NK (%) with NK cell growth. Spearman’s rank coefficient was used for correlation analysis and non-linear regression was also applied. Median values and interquartile ranges were plotted in graphs. Significance levels were set to *p* < 0.05. PB: Peripheral blood; Pre-: At seeding; Post-: After expansion.

**Table 1 ijms-24-04284-t001:** Details of the patients enrolled in this study.

Patient	Sex	Age Mean (SD) Median [Min; Max] *	Stage	Metastatic Site	Pretreatment	Duration of AIET after Surgery or the 1st Chemotherapy (Months) Mean (SD) Median [Min; Max]	Estimated Survival Prior to AIET (Months) Mean (SD) Median [Min; Max] *
		59.0 (12.4) 60.0 [32.2; 82.1]				16.2 (15.0) 12.0 [2; 48]	12.0 (4.64) 12.0 [3.0; 20.0]
PT 1	F	59.96	IV	M1 m (pleural)	Surgical/Chemo	48	12
PT 2	F	58.52	III	M0	Chemo	24	15
PT 3	M	57.56	IV	M0	Chemo	3	16
PT 4	F	61.64	III	M0	Surgical/Chemo	24	20
PT 5	F	52.47	III	M0	Surgical/Chemo	12	18
PT 6	F	59.95	IV	M1 m (bone)	Chemo	48	14
PT 7	F	59.27	IV	M0	Chemo/Radiation	2	16
PT 8	F	40.0	IV	M1 m (brain)	Chemo/Radiation	13	8
PT 9	F	51.63	IV	M1 m (brain)	Surgical/Chemo	24	12
PT 10	F	32.24	IV	M1 m	Surgical/Chemo	6	10
PT 11	M	65.76	IV	M1 m (liver)	Chemo	4	12
PT 12	F	82.12	IV	M1 m (liver)	Chemo/Radiation	12	3
PT 13	M	78.10	III	M0	Chemo/Radiation	6	8
PT 14	F	63.74	IV	M1 m (bone)	Surgical/Chemo	4	6
PT 15	M	61.52	IV	M1 m (brain)	Surgical/Chemo	13	10

* Mean, min, and max values were calculated from fifteen patients.

## Data Availability

The data that support the findings of this study are available from the corresponding author, C.T., upon reasonable request.

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
