# Peer review of "The Correlation between Peripheral Blood Index and Immune Cell Expansion in Vietnamese Elderly Lung Cancer Patients"

_ijms, 2023, doi:10.3390/ijms24054284_

Round 1
Reviewer 1 Report
The manuscript, presented by Hoang-Phuong Nguyen et al. is devoted to the search for relationships of the numbers and proportions of peripheral blood lymphocyte subpopulations ex vivo and their ability to proliferate in vitro with the aim to predict the effectivity of immune cell based therapy of cancer patients. The study includes data of 17 lung cancer patients subjected by autologous NK cell and CD8+ T cell therapy, who displayed significant variability in cell expansion. The particular question the authors addressed to this work was: “does the peripheral blood lymphocyte count associate with this expansion?” A set of data was collected on absolute counts of leukocytes and lymphocytes, percentages of CD4+, CD8+ T lymphocytes, NK cells in peripheral blood mononuclear cells, as well as on the expansion rates of CD8+ T cells and NK cells cultered using BINKIT, and a correlation analysis was performed. Several relationships were observed. Negative correlations between CD8+ T and NK cell expansion rates and ex vivo NK cell percentages and counts were emphasized by the authors.
Major concerns
The work has several limitations, some are indicated in the manuscript. The main limitation is the small size of the patient cohort, not balanced by gender, while some of the relationships observed by the authors differ for men and women.
The conclusion in the abstract is formulated incorrectly. From the data presented, only patients with lung cancer can be inferred, but not all cancer patients. It is possible that the authors would have obtained similar regularities in the study of healthy people. To find out, it would be desirable to analyze control groups. Obtaining data from a group of healthy donors, whose cells would also have been grown under similar conditions, can strengthen the work, even if it is not possible to increase the group of cancer patients.
The advisability of additional calculation of out-of-range data is not clear, especially in relation to groups of patients of different age, gender, and stage of the disease with such a small cohort size (Figure 2). The value of the information received is not obvious.
In general, the presentation of the material is rather confusing. The abstract needs to be corrected. English should also be corrected.
Minor concerns
1. Page 2, line 85: patients at stage IV are indicated twice, 4 and 11.
2. No decoding for some abbreviations (AIET, WBC) is presented at its first mention.
3. The manuscript should contain at least a brief description of the stimulation that causes lymphocytes to proliferate when using BINKIT.
4. Estimation of density of lymphocytes in numbers, G/L (Figure 1) seems strange.
5. Age and sex can hardly be considered as clinical characteristics.
6. Figure 1: the legend mentions green lines even though they are blue.
In present form the manuscript can not be published in IJMS.
Author Response
Dear Reviewer 1,
Thank you very much for your valuable comments and suggestions. Please find attached the revised manuscript entitled “The correlation of peripheral blood index to the immune cell expansion ability in Vietnamese elderly lung cancer patients”.
Kindly find below our point-to-point response highlighted in red.
Reviewer comments and suggestions:
Major concerns
- The work has several limitations, some are indicated in the manuscript. The main limitation is the small size of the patient cohort, not balanced by gender, while some of the relationships observed by the authors differ for men and women.
- We agree with the Reviewer that gender is imbalance in the patient cohort. Unfortunately, this is a retrospective study so it is difficult to recruit more patients. Therefore we have anticipated this in the manuscript and focus the analysis on the whole cohort. We have added this information in the limitation of the study.
- The conclusion in the abstract is formulated incorrectly. From the data presented, only patients with lung cancer can be inferred, but not all cancer patients. It is possible that the authors would have obtained similar regularities in the study of healthy people. To find out, it would be desirable to analyze control groups. Obtaining data from a group of healthy donors, whose cells would also have been grown under similar conditions, can strengthen the work, even if it is not possible to increase the group of cancer patients.
- Thank you for this constructive comment. We have revised the conclusion in the abstract section, specified to lung cancer: “PB indexes are intrinsically tied to immune cells health and could be leveraged to determine CD8 T and NK cells proliferation capacity for immune therapies against lung cancer”. Following the Reviewer’s recommendation, we included data on healthy donors of the same age. The data clearly showed that NK cell expansion was significantly decreased in lung cancer patients compared to healthy donors.
- The advisability of additional calculation of out-of-range data is not clear, especially in relation to groups of patients of different age, gender, and stage of the disease with such a small cohort size (Figure 2). The value of the information received is not obvious.
- We agree with the Reviewer that the small cohort size limited the comparison between groups. However, the analysis showed a significant difference in the rate of abnormal PB indexes by gender and age. For example, the out-of-range rate of CD8 T cells was observed only in males but not females, whereas the out-of-range values of NK cells h were observed only in females but not males. So, we think that these data might serve as an entry point for further studies.
- In general, the presentation of the material is rather confusing. The abstract needs to be corrected. English should also be corrected.
- Thank you for this comment. We have thoroughly reviewed the structure of the manuscript and corrected the language.
Minor concerns
- Page 2, line 85: patients at stage IV are indicated twice, 4 and 11.
- Thank you for your careful reading. The text has been modified accordingly: “The patient cohort comprised 4 patients at stage III, and 11 patients at stage IV”.
- No decoding for some abbreviations (AIET, WBC) is presented at its first mention.
- We have decoded the abbreviation AIET at the first use. For BINKIT, it is the trademark name provided by the company.
- The manuscript should contain at least a brief description of the stimulation that causes lymphocytes to proliferate when using BINKIT.
- In this study, we used BINKIT for immune cell expansion. This commercial kit was developed by Dr. Terunuma H et al. at Biotherapy Institute. The procedure of cell culture, as well as cell count, was set up following the instruction of the kit. When developing the kit, they found that during the first 3 days of culture, the number of cells should decrease due to the culture medium targeting a specific cell type, such as NK cells or CD8 T cells. It happens to most of the samples, not due to age, sex, tumor stage, type of tumor, and so on. This stage of culture they called the initial stags using the initial medium. From day 4, the medium was changed to a subculture medium to induce the proliferation of targeted cells. We have added this information in the method part.
- Estimation of density of lymphocytes in numbers, G/L (Figure 1) seems strange.
- We have changed the density of lymphocytes to cells/µL.
- Age and sex can hardly be considered clinical characteristics.
- We agree with your comment. The text has been modified accordingly. In Table 1, we just mentioned the detailed information of the patients enrolled in this study.
- Figure 1: the legend mentions green lines even though they are blue.
- Thank you for your careful reading. Figure 1 has been modified accordingly.
Sincerely yours,
Author Response
Hanoi, January 15th, 2023
Dear Reviewer 2,
Thank you very much for your valuable comments and suggestions. Please find attached the revised manuscript entitled “The correlation of peripheral blood index to the immune cell expansion ability in Vietnamese elderly lung cancer patients”.
Kindly find below our point-to-point response highlighted in red.
Reviewer comments and suggestions:
- Introduction:
Lines 65. Should the term bright be superscript?
- We thank the reviewer's comment. The term bright was superscripted.
-
- Patient characteristics.
The cohort is too heterogenous with too much variability among the different parameters (age, pretreatment, tumour stage). It does not make sense to compare different stages of lung cancer (I to IV). Moreover, the authors included 2 patients with stage I, no patients with stage II, 4 patients with stage III and 11 patients with stage IV, I think the cohort is not well balanced. The authors might focus on the late stages (III and IV), or otherwise expand the other “stage – related groups”. Besides, the authors specified their interest in the old population, but they included one 40 – year-old patient (PT10): I would suggest considering only over 60 or otherwise to compare old and young patients (under vs over 60, for example).
- We followed the reviewer’s comment and have removed the two patients at stage I to focus the analysis on the patients at stages III and IV. We also divided the cohort into two groups under and over 60 in every analysis of the manuscript.
In Table 1 I would also suggest putting the mean SD age, duration of AIET and Estimated Survival Priorr to AIET, in the same column and not at the end of the Table, it should be easier to read.
- We have arranged Table I as suggested.
What does the AIET acronym mean? Please specify.
- We have decoded the abbreviation AIET (autologous immune enhancement therapy) at the first use.
- The peripheral blood indexes
In general, it is no so clear in my mind, so I would suggest to re-write. In figure 1 I cannot see the green lines, but I see only blue lines, please revise the colours.
- The text and the figure have been modified accordingly. .
The authors declare that each data point represents on patient (Line 100), so this means the in Figure 1B, D and E, where there are 16,15 and 15 points respectively, some patients do not have Lymphocytes, or CD3, or CD4? Otherwise please explain better.
- We have added the missing data. Now all of Figure 1B, D, and E had 15 points representing 15 patients.
Line 107. How the authors calculated the percentages of out-of-range sample? I can see 5 dots outside below in panel E, 2 dots below in panel F and 3 dots below in panel G. Please clarify.
- We have reanalyzed the data in Figures 1 and 2. The out-of-range frequency was calculated by the number of samples under or upper than the normal range (corresponding to the two black dot lines in Figure 1) per the total number of patients.
Line 116-118. I think these conclusions are too trenchant with a so small cohort.
- We have rewritten the sentence to make it more consistent with the cohort in this study: “Taken together, this study demonstrated that the peripheral blood indexes of lung cancer patients enrolling in the study had a high percentage of decrease”.
- Immune cell expansion capacity
Line 128. The authors started to measure the expansion capacity of cells from day 4, might they explain why?
Line 132. Moreover, they also stated that CD8 and NK cells decrease in number during the first three days in some, but not all, patients. Might the authors explain if there is any relationship between this observation and what they wrote in Line 128? Why did they start to count cells from day 4?
Line 141. Do the authors culture the primary PBMCs for 15 days? These primary cells become old and they hardly proliferate, did the authors perform any Annexin assay to assess cell vitality after the 15 days culture?
Line 147. Might these large differences be due to age, sex, tumour stage, type of tumour and so on?
- These comments were all about the BINKIT and cell count procedure. Hence, we will give the explanation for all.
- In this study, we used BINKIT for immune cell expansion. This commercial kit was developed by Dr. Terunuma H et al. at Biotherapy Institute. The procedure of cell culture, as well as cell count, was set up following the instruction of the kit. When developing the kit, they found that during the first 3 days of culture, the number of cells should decrease due to the culture medium targeting a specific cell type, such as NK cells or CD8 T cells. It happens to most of the samples, not due to age, sex, tumor stage, type of tumor, and so on. This stage of culture they called the initial stags using the initial medium. From day 4, the medium was changed to a subculture medium to induce the proliferation of targeted cells. We have added this information in the method part.
- The immune cell expansion in out-of-range sample
This section is not clear to me, since I cannot completely understand its relevance in the general sense of the manuscript. Please clarify.
- As we mentioned in the manuscript, we observed a high percentage of lung cancer samples that have PB indexes below or above the normal range (we called them out-of-range samples). When dividing into groups by age, there was also a high difference in the rate of abnormal samples between ages above and below 60. Hence, we would like to check if there is any difference in the immune cell expansion capacity between these samples and the ones in the normal range.
Line 181. What does refer the term “post”? This term means after the expansion culture.
- We have added the explanation in the legend of Figure 3.
Do the results in Figure 5 and the relative paragraph come from observations of the results represented in Figure 1?
- Yes, the results in Figure 5 and the relative paragraph come from observations of the results represented in Figure 1.
- The relationship between the PB indexes and the immune cell expansion
Line 220-220. In this sentence, do the authors mean: The higher the percentage of CD8T and NK cells, the lower their expansion capacity?
- In our revised study, the results showed that the higher percentage of peripheral blood NK cells the lower the expansion capacity of both NK and CD8 T cells.
2.6. Discussion.
Lines 263-264. Do the authors refer to “before treatment”?
- The authors did not indicate “before or after treatment”, but emphasized the recovery of the peripheral blood lymphocytes is necessary to improve the treatment efficacy in cancer.
Lines 295-297. Do these factors (chemotherapy/radiotherapy/surgery) affect cell growth?
- We have removed the sentence mentioning the effect of treatment methods on cell growth because of the small cohort of patients.
Lines 305-315. Does this paragraph mean a reciprocal inhibitory effect on expansion by NK on CD8 and vice versa? Otherwise, it is not clear to me, please clarify?
- Yes, we observed a reciprocal inhibitory effect on expansion by NK on CD8 and vice versa. Moreover, the frequency and number of peripheral blood NK cells had an inversed correlation to NK cell expansion. We hypothesized that the high percentage of CD56dim exhaustion cells in the PB-NK population of lung cancer patients may cause this effect.
Lines 317. Chemotherapy affects the number of lymphocytes, generally, it induces their decrease. It should be taken into account.
- Yes, we agree with that. This is one of the limitations of this study since we did not analyze the influence of chemotherapy or radiotherapy on the immune cell number due to the small size cohort.
Moreover, I would suggest a minor English revision (see line 280 [had], 372 [variant; between])
- We agree with the reviewer’s comment. We revised the manuscript as reviewer’s suggestion.

Round 2
Reviewer 1 Report
In the revised version, Hoang-Phuong Nguyen et al. have improved the quality of the manuscript by making lung cancer patiens cohort more uniform and by adding comparison with healthy donors and by working on English. Still, the work still needs some additional improvements.
Major and minor concerns:
1. Incorrect sentence in the abstract. There is no the marker CD56bright, there is the marker CD56. So, cells can be CD56+ (CD56-positive), and CD56+ cells can be CD56bright (which highly express CD56) or CD56dim, they can not be CD56bright-positive.
2. Next, traditionally, CD56bright is a term designated a less differentiated NK cell subset ex vivo. It is also known, when NK cells actively proliferate, they express more CD56 per cell, but they are not fully correspond to the original CD56bright subset. Maybe better to say “NK cells with high expression level of CD56” or “NK cells highly expressed the CD56 marker”.
3. Line 109: 10 healthy individuals include 4 males and 11 females. What is meant?
4. “Density” seems to be not appropriate term for “numbers of cells” (in ml, L, or ul), because it can be considered in two ways (for example, “the density of mononuclear cells is lower than erythrocytes, that is why ficoll 1.077 can be used for isolate them). It is better to use concentration or “absolute numbers” in ml, L, or ul.
5. Figure 1 is poorly displayed, X scales of F and G is not seen.
6. Figure 1: The legend indicates that analysis was performed in PBMC, but data on white blood cells are presented. White blood cells include granulocytes that should not be in PBMC fraction.
7. Comparison of absolute numbers and proportions of lymphocytes and their subpopulations shows no significant differences between patients and healthy donors. The description of the results should be corrected accordingly.
8. Again, the advisability of presentation of out-of-range data is questionable, especially, when healthy donor cohort also contains considerable portion of out-of-range values, and the groups are small (Figure 2). These data can be presented as supplementary material or significantly reduced in main results.
9. Figure 2D: typo in “Stage”.
10. Figure 3 contains double used A, B, C, D, and the legend is only for one set.
11. The Figure 3 and other figures are blurring, it makes difficult the estimation of the results.
12. Have the authors compared statistically the data to show the difference in the NK cell expansion between lung cancer patients and healthy donors? If the difference is statistically significant, it should be indicated on the graph (expressed by stars). If not – the text describing the results should be corrected.
13. It remains not clear for me, do the correlations found between NK cell proportion and counts ex vivo and expansion rate in BINKIT are related to lung cancer patients specifically or could be observed in healthy people too. This question should be clarified or at least discussed.
14. The same cells are named PBMCs in Results and PBMNCs in Materials and Methods.
Author Response
Hanoi, January 21st, 2023
Dear Reviewer,
Thank you very much for your valuable comments and suggestions. Please find attached the revised manuscript round 2 entitled “The correlation of peripheral blood index to the immune cell expansion ability in Vietnamese elderly lung cancer patients”.
Kindly find below our point-to-point response highlighted in red.
Reviewer’s comments:
In the revised version, Hoang-Phuong Nguyen et al. have improved the quality of the manuscript by making lung cancer patiens cohort more uniform and by adding comparison with healthy donors and by working on English. Still, the work still needs some additional improvements.
Major and minor concerns:
- Incorrect sentence in the abstract. There is no the marker CD56bright, there is the marker CD56. So, cells can be CD56+ (CD56-positive), and CD56+ cells can be CD56bright (which highly express CD56) or CD56dim, they can not be CD56bright-positive.
- We have changed to “Particularly, 95% of the expanded NK cells highly expressed the CD56 marker”, Lines 24.
- Next, traditionally, CD56bright is a term designated a less differentiated NK cell subset ex vivo. It is also known, when NK cells actively proliferate, they express more CD56 per cell, but they are not fully correspond to the original CD56bright subset. Maybe better to say “NK cells with high expression level of CD56” or “NK cells highly expressed the CD56 marker”.
- We have changed to “Particularly, 95% of the expanded NK cells highly expressed the CD56 marker”, Lines 24, 25.
- Line 109: 10 healthy individuals include 4 males and 11 females. What is meant?
- We have corrected to 10 healthy individuals (6 males, 4 females), Line 88.
- “Density” seems to be not appropriate term for “numbers of cells” (in ml, L, or ul), because it can be considered in two ways (for example, “the density of mononuclear cells is lower than erythrocytes, that is why ficoll 1.077 can be used for isolate them). It is better to use concentration or “absolute numbers” in ml, L, or ul.
- We have changed to “absolute numbers” accordingly, Lines 109,110.
- Figure 1 is poorly displayed, X scales of F and G is not seen.
- We have already improved Figure 1 quality.
- Figure 1: The legend indicates that analysis was performed in PBMC, but data on white blood cells are presented. White blood cells include granulocytes that should not be in PBMC fraction.
- We have changed the Figure legend to “The peripheral blood indexes of cancer patients (n = 15) and healthy people (n=10). The complete PB cells were counted on the hematology analyzer and flow cytometry system”, Lines 105, 106.
- Comparison of absolute numbers and proportions of lymphocytes and their subpopulations shows no significant differences between patients and healthy donors. The description of the results should be corrected accordingly.
- We have corrected the comparison, by adding the phrases “but not significantly different”, in Lines 102, 104.
Again, the advisability of presentation of out-of-range data is questionable, especially, when healthy donor cohort also contains considerable portion of out-of-range values, and the groups are small (Figure 2). These data can be presented as supplementary material or significantly reduced in main results.
- We have reduced this paragraph to “However, a large portion of patients’ blood indexes was out of the normal range, in which the frequency of CD4+ T cells, CD3+ T, and lymphocytes in 60% (9/15), 53.3% (8/15) and 40% (6/15) of the patients, respectively was out of range. In healthy people, blood indexes were less likely to be out-of-range compared to lung cancer patients (Figure 1H). Taken together, these findings demonstrated that the PB indexes of lung cancer patients enrolled in the study had the tendency to be on the lower end of the spectrum and that they had a high rate of out-of-range values. Lines 115-121.
- Figure 2D: typo in “Stage”.
- We have removed this part in the Figure.
- Figure 3 contains double used A, B, C, D, and the legend is only for one set.
- We have corrected to A1, A2, B1, B2, C1, C2, D1, D2, accordingly, in Figure 3.
- The Figure 3 and other figures are blurring, it makes difficult the estimation of the results.
- We have improved the quality of these images to make them easier to analyze.
- Have the authors compared statistically the data to show the difference in the NK cell expansion between lung cancer patients and healthy donors? If the difference is statistically significant, it should be indicated on the graph (expressed by stars). If not – the text describing the results should be corrected.
- We have indicated on the graph the statistical comparison in the NK cell expansion between lung cancer patients and healthy donors in Figure 3 D2.
- It remains not clear for me, do the correlations found between NK cell proportion and counts ex vivo and expansion rate in BINKIT are related to lung cancer patients specifically or could be observed in healthy people too. This question should be clarified or at least discussed.
- We have added Figure 6 indicating the non-relationship between NK cell proportion and counts ex vivo and expansion rate in BINKIT in healthy donors, Lines 219-221. This correlation was found specifically related to lung cancer patients. We also discussed in the Discussion part, Lines 300-304.
- The same cells are named PBMCs in Results and PBMNCs in Materials and Methods.
- We have corrected to PBMCs, accordingly, in Materials and Methods.
Sincerely yours,
Reviewer 2 Report
For comments and suggestions, see the attached file "Comments for the authors_Rev2".

Author Response
Hanoi, January 21st, 2023
Dear Reviewer,
Thank you very much for your kindly evaluation and suggestions. Please find attached the revised manuscript round 2 entitled “The correlation of peripheral blood index to the immune cell expansion ability in Vietnamese elderly lung cancer patients”.
Kindly find below our point-to-point response highlighted in red.
Reviewer’s comments:
I thank the authors for the deep revision. Now the manuscript sounds better.
Anyway, I would suggest a further minor check of English spelling in the entire document.
- We have checked and corrected the English spelling in the entire document.
Moreover, in the results section, line 109, the authors described the two subject cohorts: the lung cancer cohort includes 15 patients divided into 4 males and 11 females, and the healthy donor cohort includes 10 subjects, with 4 males and 11 females, but the sum is not 10, please revise.
- We have corrected to 10 healthy individuals (6 males, 4 females), Line 88.
Sincerely yours,
Round 3
Reviewer 1 Report
I have no concerns about the revised version of the manuscript except that "in lung cancer patients " should be added in the last part of the abstract.
Author Response
Hanoi, January 25th, 2023
Dear Reviewer,
Thank you very much for your valuable suggestion. Please find attached the revised manuscript round 3 entitled “The correlation of peripheral blood index to the immune cell expansion ability in Vietnamese elderly lung cancer patients”.
Kindly find below our point-to-point response highlighted in red.
Reviewer’s comment:
I have no concerns about the revised version of the manuscript except that "in lung cancer patients " should be added in the last part of the abstract.
- We have added the phrase "in lung cancer patients" in the last part of the abstract: «(4) Conclusion: PB indexes are intrinsically tied to immune cell health and could be leveraged to determine CD8 T and NK cell proliferation capacity for immune therapies in lung cancer patients», line 29-31.
Sincerely yours,